# Bacterial Infections and Their Cell Wall Ligands Differentially Modulate Doxorubicin Sensitivity in Triple-Negative Breast Cancer Cells

**DOI:** 10.3390/microorganisms13102317

**Published:** 2025-10-07

**Authors:** Sima Kianpour Rad, Runhao Li, Kenny K. L. Yeo, Fangmeinuo Wu, Yoko Tomita, Timothy J. Price, Wendy V. Ingman, Amanda R. Townsend, Eric Smith

**Affiliations:** 1Solid Tumour Group, Basil Hetzel Institute for Translational Health Research, The Queen Elizabeth Hospital, Central Adelaide Local Health Network, Woodville South, Adelaide, SA 5011, Australia; sima.kianpourrad@adelaide.edu.au (S.K.R.); runhao.li@adelaide.edu.au (R.L.); kenny.yeo@adelaide.edu.au (K.K.L.Y.); fangmeinuo.wu@adelaide.edu.au (F.W.); yoko.tomita@sa.gov.au (Y.T.); timothy.price@sa.gov.au (T.J.P.); amanda.townsend@sa.gov.au (A.R.T.); 2Adelaide Medical School, The University of Adelaide, Adelaide, SA 5005, Australia; wendy.ingman@adelaide.edu.au; 3Medical Oncology, The Queen Elizabeth Hospital, Central Adelaide Local Health Network, Woodville South, Adelaide, SA 5011, Australia; 4Robinson Research Institute, The University of Adelaide, Adelaide, SA 5005, Australia; 5Discipline of Surgery, The University of Adelaide, Adelaide, SA 5005, Australia

**Keywords:** triple-negative breast cancer (TNBC), intratumoral microbiome, *Pseudomonas aeruginosa*, *Staphylococcus aureus*, intracellular bacterial persistence, doxorubicin, lipopolysaccharide (LPS), lipoteichoic acid (LTA), toll-like receptors (TLR2; TLR4), chemotherapy sensitization

## Abstract

Background: Triple-negative breast cancer (TNBC) is an aggressive subtype with limited treatment options and poor clinical outcomes. Emerging evidence suggests that the tumor-associated microbiome may influence disease progression and therapy response. Methods: We investigated how the Gram-negative bacterium *Pseudomonas aeruginosa* and Gram-positive bacterium *Staphylococcus aureus*, together with their cell wall components lipopolysaccharide (LPS) and lipoteichoic acid (LTA), modulate doxorubicin (DOX) efficacy in TNBC cells. Using gentamicin protection combined with flow cytometry of eFluor 450-labeled bacteria and CFU quantification, we assessed bacterial uptake, persistence, and effects on drug response in MDA-MB-468, MDA-MB-231, and MDA-MB-453 cells. Results: Both bacteria entered TNBC cells and survived for several days in a cell line-dependent manner. Notably, bacterial infection and purified cell wall ligands (LPS and LTA) significantly increased DOX accumulation and enhanced cytotoxicity in MDA-MB-468 and MDA-MB-231, but not in MDA-MB-453. The similar effects of LPS and LTA implicate Toll-like receptor signaling (TLR2 and TLR4) in modulating drug uptake. Conclusions: These findings demonstrate that bacterial infection and associated ligands can enhance doxorubicin uptake and cytotoxicity in TNBC cells, implicating TLR signaling as a potential contributor. Our results highlight the importance of host–microbe interactions in shaping chemotherapy response and warrant further investigation into their therapeutic relevance.

## 1. Introduction

Triple-negative breast cancer (TNBC) is a clinically aggressive subtype defined by the absence of estrogen receptor (ER), progesterone receptor (PR), and HER2 expression, leaving few targeted therapeutic options. Systemic chemotherapy remains the mainstay of treatment, with anthracyclines like doxorubicin being commonly used. However, the therapeutic efficacy of doxorubicin is often compromised by the development of chemoresistance and dose-limiting toxicity [1]. Although the overexpression of ATP-binding cassette (ABC) efflux transporters, particularly ABCB1 (P-glycoprotein), is a well-characterized mechanism of resistance, emerging evidence, including our own, suggests that additional, ABCB1-independent mechanisms also impair drug accumulation in TNBC cells [2].

Recent research has revealed that the tumor microenvironment plays a critical role in cancer progression and therapeutic responses. Among its many components, the intratumoral microbiome is increasingly recognized as a significant factor in modulating these processes, including in breast cancer [3,4,5,6,7]. Our recent meta-analysis of 11 studies encompassing 1260 fresh breast tissue specimens analyzed by 16S rRNA sequencing identified *Staphylococcus* and *Pseudomonas* as two of the most abundant and prevalent genera [8]. Notably, these genera belong to distinct taxonomic groups—*Staphylococcus* being Gram-positive and *Pseudomonas* Gram-negative—implying potentially divergent host interactions.

Some intratumoral bacteria, including *S. aureus*, can internalize into cancer cells, and we have previously shown that viable *S. aureus* infection upregulates PD-L1 expression via TLR2 signaling, suggesting its role in immune evasion [9]. Other studies have demonstrated that bacteria can modulate host pathways, including those involved in chemotherapy resistance [10,11,12,13]. However, while *Pseudomonas aeruginosa* is known to secrete factors like quorum-sensing molecules that influence breast cancer cell behavior, its ability to enter and persist in TNBC cells and its effects on drug accumulation and cytotoxicity remain unexplored [14].

Bacterial structural components, such as lipoteichoic acid (LTA) from Gram-positive bacteria and lipopolysaccharide (LPS) from Gram-negative species, interact with host cells and activate Toll-like receptors (TLRs), notably TLR2 and TLR4. These interactions have been shown to impact drug sensitivity in cancer models [15,16,17,18]. However, the role of these bacterial ligands in modulating doxorubicin accumulation or cytotoxicity in TNBC cells remains unclear.

Given these findings, we hypothesized that intracellular *S. aureus* and *P. aeruginosa*, along with their ligands, may modulate doxorubicin accumulation and cytotoxic response in TNBC cell lines, contributing to variability in treatment efficacy. In this study, we examined the internalization, persistence, and functional effects of *P. aeruginosa* in three molecularly distinct TNBC cell lines: MDA-MB-468 (basal-like 1), MDA-MB-231 (mesenchymal stem-like), and MDA-MB-453 (luminal androgen receptor subtype). Building on our prior work with *S. aureus* [9], we compared the effects of viable intracellular *S. aureus* and *P. aeruginosa*, their exotoxins and their respective cell wall components (LTA and LPS) on doxorubicin accumulation and cytotoxicity.

This study identifies intracellular bacteria as modulators of doxorubicin response in TNBC and highlights microbial factors as underappreciated contributors to the heterogeneity of chemotherapy outcomes.

## 2. Materials and Methods

### 2.1. Reagents

Purified lipoteichoic acid (LTA) from *S. aureus* (InvivoGen, San Diego, CA, USA; Cat# tlrl-pslta) and lipopolysaccharide (LPS) from *P. aeruginosa* (Sigma-Aldrich, St. Louis, MO, USA; Cat# L9143) were used as representative bacterial cell wall components. Doxorubicin hydrochloride was obtained from Pfizer Australia Pty Ltd. (Sydney, NSW, Australia).

### 2.2. Breast Cell Lines

Human TNBC cell lines MDA-MB-231, MDA-MB-468, and MDA-MB-453 were acquired from the American Type Culture Collection (ATCC, Manassas, VA, USA). Cells were cultured in Dulbecco’s Modified Eagle’s Medium (DMEM; Gibco, Thermo Fisher Scientific, Waltham, MA, USA; Cat# 12430062), supplemented with 10% heat-inactivated fetal bovine serum (FBS; Gibco; Cat# 26140079) and 1% penicillin-streptomycin (Gibco; Cat# 15140122). Cells were maintained at 37 °C in a humidified atmosphere containing 5% CO_2_. Mycoplasma contamination was ruled out using the MycoAlert Mycoplasma Detection Kit (Lonza, Basel, Switzerland; Cat# LT07-318) [9].

### 2.3. Bacterial Culture and Fluorescent Labeling

*Pseudomonas aeruginosa* (Schroeter) Migula ATCC 15692 was streaked onto Luria–Bertani Agar (LBA; Oxoid, Thermo Fisher Scientific; Cat# 10855001) and incubated at 37 °C overnight. Colonies (3–4) were then transferred into 35 mL of Luria–Bertani (LB) broth in 50 mL conical tubes and cultured overnight at 37 °C under aerobic conditions with agitation (180 rpm).

The *Staphylococcus aureus* subsp. *aureus* Rosenbach ATCC 25923 reference strain was first cultured on Tryptone Soya Agar (Oxoid, Thermo Fisher Scientific; Cat# CM0131). Individual colonies were then transferred into 35 mL of Tryptone Soya Broth (Oxoid, Thermo Fisher Scientific, Cat# CM0129) in 50 mL conical tubes and incubated overnight at 37 °C with shaking at 180 rpm under aerobic conditions.

For fluorescent labeling, overnight cultures of both *S. aureus* and *P. aeruginosa* were standardized to a concentration of 1.0 × 10^9^ CFU/mL, pelleted by centrifugation at 3200× *g* for 10 min, and washed three times with sterile, protein-free phosphate-buffered saline (PBS). The bacterial pellet was then resuspended in 750 µL of 10 µM eBioscience Cell Proliferation Dye eFluor 450 (Invitrogen, Thermo Fisher Scientific, San Diego, CA, USA; Cat# 65-0842-90), following the manufacturer’s protocol. The suspension was incubated at 37 °C in the dark for 30 min with gentle agitation. Excess dye was quenched by incubating the bacteria in DMEM supplemented with 10% FBS for 10 min, followed by two additional PBS washes.

For infection assays, fluorescently labeled suspensions of *S. aureus* and *P. aeruginosa* were adjusted to McFarland standard 0.5 (approximately 1.5 × 10^8^ CFU/mL) and diluted in DMEM to achieve the appropriate multiplicity of infection (MOI) before being applied to mammalian cell cultures. These labeled bacteria were then used in cell-based assays to evaluate bacterial uptake and intracellular persistence. While data for *S. aureus* are presented [9], the current study focuses on *P. aeruginosa* and offers comparative insights between the two species.

### 2.4. Gentamicin Protection Assay

Breast cancer cell lines were seeded at a density of 1.5 × 10^5^ cells per well in 12-well plates and cultured overnight at 37 °C in a humidified incubator with 5% CO_2_. The following day, cells were washed with sterile Dulbecco’s Phosphate-Buffered Saline (DPBS) (Gibco, Thermo Fisher Scientific; Cat# 14190144) and infected with either eFluor 450-labeled or unlabeled *S. aureus* or *P. aeruginosa* in antibiotic-free DMEM containing 10% FBS. Bacteria were applied at multiplicities of infection (MOI) of 10, 50, or 100 and incubated for 2 h at 37 °C in the dark to prevent photobleaching.

After infection, cells were washed thoroughly with DPBS and incubated in DMEM containing 200 µg/mL gentamicin (Gibco, Thermo Fisher Scientific; Cat# 15750060) for 1 h to eliminate extracellular bacteria. Cells were then washed with DPBS and maintained DMEM supplemented with 50 µg/mL gentamicin for 24 h. For extended time points, *S. aureus*-infected cells were maintained in 5 µg/mL gentamicin, whereas *P. aeruginosa*-infected cells were maintained in 50 µg/mL gentamicin, consistent with the MIC determined for *P. aeruginosa*.

### 2.5. Measurement of P. aeruginosa Internalization by Flow Cytometry

Breast cancer cell lines were infected with either unlabeled or eFluor 450-labeled *P. aeruginosa* at a maximum MOI of 50, based on our optimization showing that higher bacterial loads were cytotoxic across all cell lines. At 24-h post-infection, cells were detached using TrypLE Express (Gibco; Cat# 12604021), centrifuged at 300× *g* for 5 min, and washed twice with DPBS. To assess cell viability, 100 µL of ViaDye Red Fixable Viability Dye (Cytek Biosciences; Cat# R7-60008; 1:100,000 dilution in protein-free PBS) was added, followed by 20 min of incubation in the dark and two additional washes. A minimum of 30,000 single-cell events per sample were acquired using a Cytek Aurora spectral flow cytometer (Cytek Biosciences, Fremont, CA, USA), and dead cells were excluded based on ViaDye positivity. The proportion of *P. aeruginosa*-infected cells was determined by comparing eFluor 450 fluorescence to that of cells infected with unlabeled bacteria, and intracellular bacterial load was quantified using the geometric mean fluorescence intensity (GMFI). Data analysis was performed using FlowJo v10.10.0 (BD Biosciences).

### 2.6. Clearance of Viable Intracellular P. aeruginosa

To assess bacterial persistence, breast cancer cell lines infected with *P. aeruginosa* at an MOI of 50 were harvested at 0-, 1-, 3-, and 15-days post-infection. A total of 3 × 10^4^ cells were lysed in 200 µL of 1% Triton X-100 for 5 min to release intracellular bacteria. The resulting lysates were serially diluted in sterile PBS and spread onto LBA plates, followed by incubation at 37 °C for 24 h. Colony-forming units (CFUs) were enumerated after incubation, and bacterial load was expressed as CFU/mL using the equation: CFU/mL = (number of colonies × dilution factor)/volume plated (mL). At each time point, uninfected lysed cells served as negative controls to account for background contamination.

### 2.7. Doxorubicin Accumulation Assays

Infected cells with *S. aureus* were exposed at MOIs of 10 and 50 for MDA-MB-468, and MOIs of 50 and 200 for MDA-MB-231 and MDA-MB-453. In parallel, *P. aeruginosa* was used at MOIs of 10 and 50 across all three cell lines. After infection and removal of extracellular bacteria, the cells were treated with 100 nM doxorubicin and cultured in media containing 50 µg/mL gentamicin for 24 h [9].

Following this incubation, both adherent and detached cells were collected for flow cytometric analysis. Cells were stained with ViaDye Red Fixable Viability Dye (Cytek Biosciences) to exclude non-viable populations, and intracellular doxorubicin accumulation in viable cells was quantified based on the intrinsic fluorescence of doxorubicin using a Cytek Aurora spectral flow cytometer (Cytek Biosciences, Fremont, CA, USA). A minimum of 30,000 single-cell events were acquired per sample, and data were analyzed using FlowJo software (version 10.10.0; BD Biosciences).

The same procedure was applied to cells treated with 10 µg/mL LTA or 10 µg/mL LPS, together with 100 nM doxorubicin, to assess the effects of bacterial cell wall ligands on drug accumulation. In addition, *S. aureus* exotoxins (tested at 1 µg/mL and at their IC_50_ concentration of 10 µg/mL) and *P. aeruginosa* exotoxins (1 µg/mL and IC_50_ of 8 µg/mL) were evaluated for their influence on intracellular doxorubicin accumulation.

### 2.8. Growth Inhibition Assays

Infected and uninfected TNBC cells were seeded at a density of 3.3 × 10^3^ cells per well in 96-well flat-bottom plates (Corning; Cat# 3599) and incubated overnight. The following day, infected cells were treated with doxorubicin at their respective IC_50_ concentrations: 29.2 nM for MDA-MB-468, 37.7 nM for MDA-MB-231, and 51 nM for MDA-MB-453) [2]. In parallel, uninfected cells were treated with LTA or LPS at 1, 10, and 100 µg/mL, either alone or in combination with doxorubicin at the IC_50_ concentration for each cell line.

After 5 days of treatment, cell growth was quantified using a crystal violet staining assay, as previously described [9], and expressed relative to uninfected controls (for bacterial infection) or untreated controls (for ligand treatment).

### 2.9. Statistical Analysis

Statistical analyses were performed using Prism 10 for macOS (Version 10.6.0 (796), 12 August 2025; GraphPad Software Inc., La Jolla, CA, USA).

## 3. Results

### 3.1. Pseudomonas aeruginosa Internalization Varies Between TNBC Cell Lines, Distinct from Staphylococcus aureus Uptake Patterns

We assessed the internalization of eFluor 450-labeled *P. aeruginosa* in viable MDA-MB-468, MDA-MB-231, and MDA-MB-453 cells using spectral flow cytometry with a gentamicin protection assay (Figure 1A). Infections at multiplicities of infection (MOIs) of 5, 10, and 50 revealed clear, dose-dependent, cell line-specific differences in both the proportion of positive cells (Figure 1B) and bacterial load, expressed as geometric mean fluorescence intensity (GMFI; Figure 1C). One-way ANOVA confirmed significant effects of MOI on the percentage of positive cells in MDA-MB-231 (F(2,6) = 67.03, *p* < 0.0001) and MDA-MB-468 (F(2,6) = 36.35, *p* = 0.0004), both showing strong linear trends across increasing MOIs (*p* < 0.01). By contrast, MDA-MB-453 displayed no significant MOI-dependent changes (F(2,6) = 1.10, *p* = 0.39), consistent with its minimal uptake. Analysis of GMFI revealed parallel results, with highly significant MOI effects in MDA-MB-231 (F(2,6) = 427.7, *p* < 0.0001) and MDA-MB-468 (F(2,6) = 297.6, *p* < 0.0001), confirming robust dose-dependent increases in bacterial load, whereas MDA-MB-453 exhibited negligible changes. Internalization correlated strongly with bacterial load across MOIs in MDA-MB-231 and MDA-MB-468 (R^2^ = 0.96–0.99, *p* < 0.0001), whereas MDA-MB-453 showed no meaningful correlation (Figure 1D).

Having established MOI-dependent effects within each cell line, we next compared internalization between the three TNBC models. MDA-MB-231 cells were most susceptible across all MOIs, with 30 ± 3% of viable cells positive at MOI 50 and the highest GMFI (32,459 ± 2200). MDA-MB-468 showed moderate uptake (12 ± 2.4%; GMFI 4119 ± 58), whereas MDA-MB-453 exhibited negligible internalization (0.25 ± 0.09%; GMFI 150 ± 25.2). Both the proportion of positive cells and the bacterial load differed significantly among cell lines at each MOI.

Although labeling efficiency was lower for *P. aeruginosa* than for *S. aureus*
Appendix A, the method confirmed intracellular presence and enabled cross-cell line comparisons. We compared the *P. aeruginosa* findings with our previously reported *S. aureus* data [9]. A direct comparison at MOI 50 (Table 1) showed distinct uptake patterns: *P. aeruginosa* internalization was highest in MDA-MB-231, whereas *S. aureus* was most efficiently internalized by MDA-MB-468, with limited uptake in MDA-MB-453 for both bacteria.

### 3.2. Intracellular P. aeruginosa Persistence Differs Between TNBC Cell Lines

To assess intracellular persistence, we quantified viable intracellular *P. aeruginosa* by colony-forming unit (CFU) enumeration at serial points following infection at MOI 50 (Figure 2 and Appendix A). At Day 0, consistent with the flow cytometry data, more *P. aeruginosa* were internalized by MDA-MB-231 than by MDA-MB-468, whereas MDA-MB-453 remained largely resistant. By Day 1, viable bacteria were still present in both MDA-MB-231 and MDA-MB-468, with higher recovery from MDA-MB-231, while no colonies were detected from MDA-MB-453. Two-way ANOVA confirmed a significant effect of cell line (*p* < 0.001), supporting the flow cytometry findings.

To compare early intracellular clearance kinetics, we regressed log_10_(CFU + 1) on time using data from Days 0–3. MDA-MB-231 showed a downward trend that did not reach statistical significance (slope –0.72 [95% CI: –1.78 to 0.35] log_10_ CFU/day; *p* = 0.13), whereas MDA-MB-468 exhibited a significant decline (slope –0.72 [95% CI: –1.26 to –0.17] log_10_ CFU/day; *p* = 0.02). The slope comparison test confirmed no significant difference between the two lines (*p* = 0.12). Despite similar clearance rates, MDA-MB-231 maintained a substantially higher intracellular burden across this interval (main effect of cell line *p* < 0.001). MDA-MB-453 showed negligible recovery at all time points, with no measurable decline (slope –0.03 [95% CI: –0.10 to 0.04] log_10_ CFU/day, *p* = 0.34). Robust regression (Theil–Sen estimator) produced comparable slope estimates (–0.50 for MDA-MB-231, –0.66 for MDA-MB-468, ≈0.00 for MDA-MB-453).

Importantly, viable *P. aeruginosa* were still recovered from MDA-MB-231 at Day 15, whereas longer-term CFU data were not collected for MDA-MB-468 or MDA-MB-453. Thus, only MDA-MB-231 can be confirmed to support prolonged bacterial persistence.

Together, these results demonstrate that MDA-MB-231 and MDA-MB-468 reduce intracellular *P. aeruginosa* at similar early rates, but MDA-MB-231 sustains a markedly higher bacterial burden and uniquely permits long-term bacterial survival, while MDA-MB-453 remains resistant to internalization.

### 3.3. Acute Cytotoxicity Induced By P. aeruginosa Infection is Cell Line-Dependent

We next evaluated the acute cytotoxicity of *P. aeruginosa* by assessing viable cell counts 2 h post-infection (Figure 3A) At MOI 100, all three TNBC cell lines showed significant reductions in viability (*p* < 0.0001): MDA-MB-468 exhibited the greatest decrease (88 ± 2.5% reduction relative to uninfected controls), followed by MDA-MB-231 (82.3 ± 3.3%) and MDA-MB-453 (78.2 ± 0%). At MOI 50, viability decreased by 24.4 ± 4.3% in MDA-MB-468, 38.8 ± 2.4% in MDA-MB-231 and 23.3 ± 5.8% in MDA-MB-453. MOI 10 had no significant impact on viability.

We also examined the effects of *P. aeruginosa* exotoxins (Figure 3B). Exotoxin exposure reduced cell viability in a dose-dependent manner, with IC_50_ values of 3.2 µg/mL (95% CI: 3–3.6 µg/mL) for MDA-MB-468, 7.8 µg/mL (95% CI: 7.6–8.0 µg/mL) for MDA-MB-231, and 14.7 (95% CI: 14.09–15.33 µg/mL) for MDA-MB-453, relative to untreated controls.

### 3.4. Bacterial Infection Enhances Doxorubicin Accumulation in TNBC Cells

Having established that *S. aureus* and *P. aeruginosa* exert acute, MOI-dependent cytotoxic effects on TNBC cells, we next asked whether bacterial infection influences doxorubicin accumulation. To ensure that bacterial viability did not confound these assays, we first confirmed that doxorubicin had no effect on the growth of either species in minimal inhibitory concentration (MIC) assays Appendix A.

TNBC cells were then infected with *S. aureus* or *P. aeruginosa* at different MOIs and subsequently treated with 100 nM doxorubicin for 24 h (Figure 4 and Appendix A). In uninfected controls doxorubicin accumulation (quantified by flow cytometry based on the intrinsic fluorescence of doxorubicin) differed by cell line, with MDA-MB-468 > MDA-MB-231 > MDA-MB-453 (54,427 ± 393; 43,694 ± 1876; 34,652 ± 1512, respectively).

The MOIs used for *S. aureus* were selected based on our previous study, where internalization was confirmed and conditions were optimized to achieve maximal uptake while maintaining the highest possible cell viability [9]. For *P. aeruginosa*, MOIs were similarly optimized in the present study to balance bacterial uptake with cell survival.

*S. aureus* infection significantly increased doxorubicin accumulation in MDA-MB-468 and MDA-MB-231, but not in MDA-MB-453 (Figure 4A). In MDA-MB-468, GMFI rose to 61,984 ± 647 at MOI 10 (1.14-fold, *p* < 0.01) and 66,652 ± 2648 at MOI 50 (1.22-fold, *p* < 0.001), with MOI 50 significantly greater than MOI 10 (*p* < 0.05), one-way ANOVA confirmed a significant effect of bacterial treatment on doxorubicin accumulation (F(2,6) = 45.17, *p* = 0.0002). In MDA-MB-231, GMFI increased to 55,123 ± 758 at MOI 50 (1.26-fold, *p* < 0.0001) and 65,752 ± 573 at MOI 200 (1.50-fold, *p* < 0.0001), with MOI 200 significantly greater than MOI 50 (*p* < 0.0001), one-way ANOVA again confirmed a significant treatment effect (F(2,6) = 58.41, *p* = 0.0001). By contrast, MDA-MB-453 showed no significant change at any MOI.

*P. aeruginosa* infection also increased doxorubicin accumulation in MDA-MB-468 and MDA-MB-231, but not in MDA-MB-453 (Figure 4B). In MDA-MB-468, GMFI increased to 57,390 ± 1617 at MOI 10 (1.05-fold, *p* < 0.01) and to 62,742 ± 754 at MOI 50 (1.15-fold, *p* < 0.001), with MOI 50 significantly greater than MOI 10 (*p* < 0.001), one-way ANOVA confirmed a significant overall effect of MOI (F(2,6) = 1.47, *p* = 0.30. In MDA-MB-231, GMFI rose to 50,507 ± 1312 at MOI 10 (1.09-fold, *p* < 0.01) and 55,477 ± 221 at MOI 50 (1.20-fold, *p* < 0.0001), with MOI 50 significantly greater than MOI 10 (*p* < 0.05), one-way ANOVA confirmed strong MOI-dependent effects (F(2,6) = 15.95, *p* = 0.0048). No significant changes were observed in MDA-MB-453.

Together, these findings demonstrate that bacterial infection enhances doxorubicin accumulation in TNBC cells in a species-, cell line-, and MOI-dependent manner, with the strongest effects observed in cell lines that support higher bacterial uptake and persistence.

### 3.5. LTA and LPS Increase Doxorubicin Accumulation, but Exotoxins Have No Effect

Since viable bacterial infection enhanced doxorubicin accumulation, we next examined whether bacterial cell wall ligands and exotoxins could produce similar effects. The impact of *S. aureus* lipoteichoic acid (LTA) and *P. aeruginosa* lipopolysaccharide (LPS) on intracellular doxorubicin accumulation is shown in Figure 5A.

In MDA-MB-468, both LTA and LPS significantly increased doxorubicin accumulation compared with doxorubicin alone. LTA enhanced uptake by 1.2-fold (*p* < 0.001; 57,156 ± 1200 vs. 69,474 ± 720 GMFI), while LPS produced a 1.1-fold increase (*p* < 0.01; 57,156 ± 1200 vs. 65,409 ± 800 GMFI). One-way ANOVA confirmed a significant overall effect of treatment (F(2,6) = 29.11, *p* = 0.0008). Similarly, in MDA-MB-231, LTA and LPS also elevated doxorubicin accumulation relative to doxorubicin alone. LTA increased GMFI by ~1.2-fold (*p* < 0.01; 48,300 ± 1050 vs. 58,250 ± 900), and LPS by ~1.1-fold (*p* < 0.05; 48,300 ± 1050 vs. 53,600 ± 750). One-way ANOVA likewise showed a significant treatment effect (F(2,6) = 25.87, *p* = 0.0011).

In contrast, neither ligand significantly altered doxorubicin accumulation in MDA-MB-453 cells, consistent with results from viable bacterial infections.

In contrast, neither 1 µg/mL nor the IC_50_ concentrations of exotoxins (*S. aureus*: 10 µg/mL [9]; *P. aeruginosa*: 8 µg/mL) significantly altered doxorubicin accumulation in MDA-MB-231 treated cells (Figure 5B).

### 3.6. Growth-Inhibitory Effects of Intracellular S. aureus and P. aeruginosa and Cell Wall Components, Alone and in Combination with Doxorubicin, in TNBC Cells

We next investigated whether live bacteria and their cell wall ligands modulate doxorubicin sensitivity in TNBC cells over a 5-day period(Figure 6).

In MDA-MB-468 cells, *S. aureus* infection at MOI 10 or 50 did not affect growth, whereas *P. aeruginosa* at MOI 50 significantly reduced growth (Figure 6A). Doxorubicin alone at its IC_50_ concentration (29.2 nM; Appendix A) decreased proliferation by ~50% as expected, and co-treatment with either bacterium produced no additional change.

In MDA-MB-231 cells, *S. aureus* reduced proliferation only at high MOI (200), while *P. aeruginosa* had no effect (Figure 6D). Doxorubicin at its IC_50_ concentration (37.7 nM [2]), reduced growth to 58 ± 2%, and this inhibition was significantly enhanced by *S. aureus* at MOI 200, but not by *P. aeruginosa.*

We next examined the effects of bacterial ligands, either alone or in combination with IC_50_ concentrations of doxorubicin. In all three TNBC lines, LPS (Figure 6B,E,G) and LTA (Figure 6C,F,H) alone did not alter growth at the concentrations tested (1–100 µg/mL). However, when combined with doxorubicin at IC_50_ concentrations, both ligands significantly enhanced doxorubicin-induced growth inhibition in MDA-MB-468 and MDA-MB-231 cells in a concentration-dependent manner. In MDA-MB-468 cells, growth declined to ~40% with LPS and ~30% with LTA at the highest concentrations tested (Figure 6B,C). Comparable effects were observed in MDA-MB-231 cells, where growth decreased to ~45% and ~34%, respectively (Figure 6E,F). By contrast, MDA-MB-453 cells showed no change in response to LPS or LTA either alone or in combination with doxorubicin at its IC_50_ concentration (51 nM [2]).

Together, these findings demonstrate that while bacterial ligands alone do not impair TNBC cell growth, they significantly enhance doxorubicin-induced growth inhibition in MDA-MB-468 and MDA-MB-231 cells in a concentration-dependent manner, consistent with their ability to increase intracellular doxorubicin accumulation.

## 4. Discussion

This study provides new evidence that *Pseudomonas aeruginosa* can invade and persist within triple-negative breast cancer (TNBC) cells, extending our earlier findings with *Staphylococcus aureus* [9]. Using complementary approaches (gentamicin protection with flow cytometry of eFluor 450-labeled bacteria and CFU quantification), we confirmed that *P. aeruginosa* internalizes into MDA-MB-231 and MDA-MB-468 cells, remaining viable for several days, with prolonged persistence confirmed only in MDA-MB-231. In contrast, MDA-MB-453 showed minimal uptake and negligible recovery, consistent with relative resistance to *P. aeruginosa* internalization. These observations highlight that intracellular persistence is both bacterial species- and cell line dependent, reflecting intrinsic differences in host defence pathways [9].

Despite evidence of *P. aeruginosa* persistence only in MDA-MB-231 cells, cytotoxicity was largely attributable to acute, MOI-dependent effects on viability across all three TNBC lines. A similar response was observed with *P. aeruginosa* exotoxins, consistent with type III secretion system-mediated toxicity [19,20,21,22,23]. MDA-MB-231 cells tolerated prolonged intracellular colonization, regaining proliferative capacity despite harbouring viable bacteria, suggesting that acute toxin exposure, rather than persistence, is the dominant driver of growth inhibition.

A key novel finding is that both *S. aureus* and *P. aeruginosa* infection enhanced doxorubicin (DOX) accumulation in MDA-MB-231 and MDA-MB-468 cells, but not in MDA-MB-453, leading to sustained reductions in growth. This effect cannot be explained solely by intracellular persistence: our previous work showed that MDA-MB-453 can harbor viable *S. aureus* at MOI 50 up to Day 2 [9], yet infection in this line did not alter DOX accumulation. These results instead indicate that bacterial infection and the release of structural ligands, rather than persistence alone, are the key drivers of enhanced drug uptake. Purified cell wall ligands (LTA and LPS) reproduced this effect and potentiated DOX-induced growth inhibition in a dose-dependent manner, whereas crude exotoxin preparations strongly inhibited proliferation without increasing DOX accumulation. Thus, structural components of the bacterial cell wall, rather than secreted toxins, are primarily responsible for promoting drug uptake. This sensitizing effect was dependent on bacterial species, MOI, and cell line, and is consistent with recent reports showing that both intestinal and intratumoral microbiota can regulate doxorubicin responsiveness and therapy outcomes [24,25].

These findings support a model in which bacterial surface molecules enhance chemotherapy response by altering host cell membrane properties or signaling pathways [26,27]. Engagement of pattern recognition receptors such as TLR2 (for LTA) and TLR4 (for LPS) may play a central role, consistent with studies in cardiomyocytes, intestinal epithelial cells, and breast cancer lines showing that TLR activation modulates DOX uptake and toxicity [28,29,30,31,32]. One possible mechanism is crosstalk between TLR signaling and ABC transporters, key mediators of multidrug resistance in TNBC [33]. Although our previous work showed that DOX accumulation increased independently of ABCB1 transcript levels, accumulating evidence indicates that TLR ligands can regulate transporter expression and activity via inflammatory pathways [34,35,36]. Future studies should determine whether bacterial ligands directly modulate ABC transporter function in TNBC cells.

The resistant phenotype of MDA-MB-453 across assays further highlights the potential importance of cellular context in shaping host–microbe interactions. Differences in innate receptor expression, immune signaling capacity, or transporter activity may underlie the divergent responses among TNBC subtypes [37].

This study has several limitations. All experiments were performed in vitro using established TNBC cell lines, which do not fully capture the complexity of in vivo tumor–microbiome–immune interactions. Although we examined three TNBC lines representing distinct subtypes, this does not reflect the full heterogeneity of the disease. Mechanistic links to TLR2, TLR4, and ABC transporters were inferred from ligand responsiveness and prior literature but were not directly tested by inhibition or genetic knockdown. Exotoxins were assessed as crude mixtures, so the contribution of individual virulence factors was not resolved. Finally, the bacterial doses and ligand concentrations used may not precisely mirror intratumoral exposure in patients. These limitations should be addressed in future studies using additional models, including in vivo systems, to validate the clinical relevance of our findings. In particular, the immunomodulatory effects of LPS and LTA through TLR2/TLR4 signaling may further potentiate anti-tumor responses when combined with doxorubicin, but their established toxicities highlight the need to explore safer TLR agonists (e.g., Pam3CSK4, MPLA) or delivery systems as alternatives.

These results also have translational implications. Modulating tumor-associated bacteria or their products could represent a novel strategy to enhance chemotherapy response in TNBC. While antibiotics remain the standard approach to bacterial elimination, broad-spectrum regimens can disrupt commensal microbiota and influence systemic therapy responses [38]. Selective microbiome-targeted approaches, or strategies harnessing bacterial components to potentiate drug accumulation, may offer complementary routes to overcome chemoresistance in aggressive breast cancers [39].

## 5. Conclusions

Our study demonstrates that both *P. aeruginosa* and *S. aureus* infection, together with their major cell wall components LPS and LTA, but not their exotoxins, enhance doxorubicin accumulation and potentiate its antiproliferative effects in TNBC cells. These findings reveal a previously unrecognized role for bacterial infection and associated ligands in sensitizing breast cancer cells to chemotherapy. Further mechanistic studies should explore how TLR2 and TLR4 signaling influences drug uptake and retention. Collectively, our work highlights the importance of host–microbe interactions in shaping therapeutic outcomes and provides a rationale for microbiome-informed strategies to improve TNBC treatment. These results, however, are limited to in vitro models and should be validated in vivo to establish their clinical relevance.

## Figures and Tables

**Figure 1 microorganisms-13-02317-f001:**
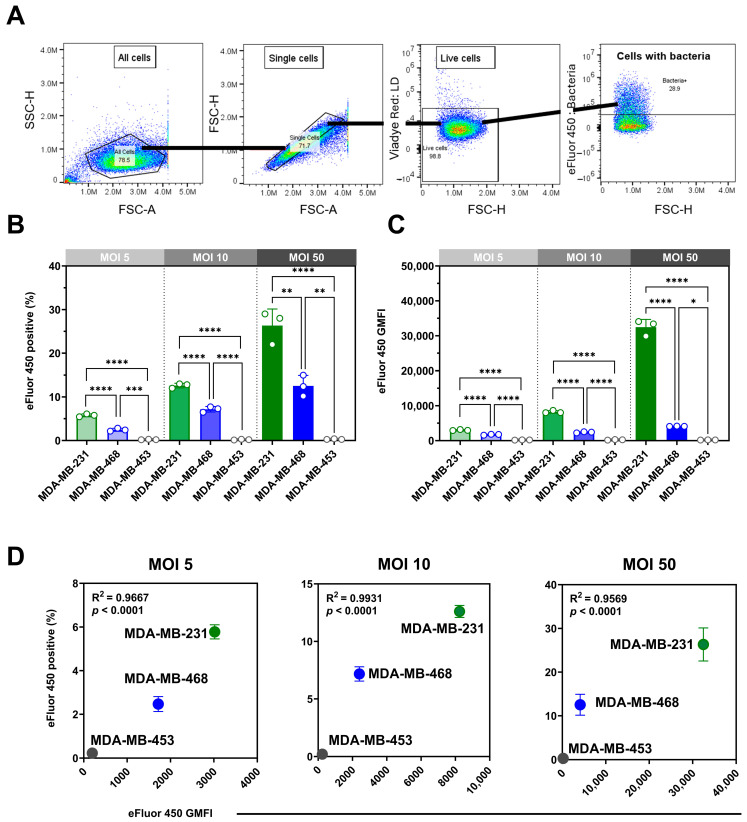
Quantification of *Pseudomonas aeruginosa* internalization in TNBC cell lines by spectral flow cytometry. (**A**) Representative gating strategy: cells were sequentially gated by size (FSC/SSC), singlets, and viability (ViaDye Red exclusion), followed by detection of eFluor 450-positive cells indicating intracellular *P. aeruginosa*. TNBC cells were incubated with eFluor 450-labelled bacteria for 2 h at multiplicities of infection (MOI 5, 10, 50), washed, cultured in 200 µg/mL gentamicin for 1 h, then maintained in 50 µg/mL gentamicin for 24 h to eliminate extracellular bacteria prior to analysis. (**B**) Percentage of eFluor 450-positive cells across TNBC lines, showing cell line-dependent differences in internalization. (**C**) Geometric mean fluorescence intensity (GMFI) of eFluor 450 in infected cells, reflecting bacterial load. (**D**) Correlation between the percentage of eFluor 450-positive cells and bacterial load (GMFI) across MOIs, confirming the dose-dependent relationship. Data are presented as mean ± SD from *n* = 3 independent experiments (each performed in triplicate wells, averaged per experiment). For (**B**,**C**), MOI effects were tested within each cell line using one-way ANOVA with polynomial trend analysis; exact F-statistics, degrees of freedom, and *p* values are reported in the Results. Asterisks in (**B**,**C**) denote significant differences between cell lines at a fixed MOI (one-way ANOVA with Tukey’s post hoc test). For (**D**), correlations were evaluated by Pearson’s regression (R^2^ = 0.96–0.99, *p* < 0.0001). (* *p* < 0.05, ** *p* < 0.01; *** *p* < 0.001; **** *p* < 0.0001).

**Figure 2 microorganisms-13-02317-f002:**
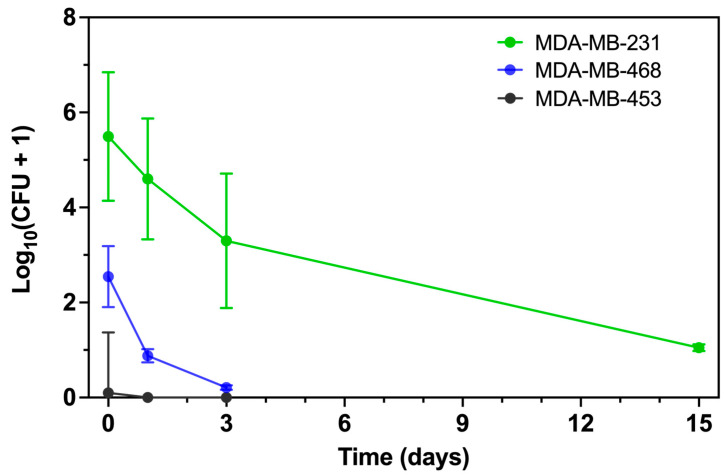
Intracellular persistence of *P. aeruginosa* in TNBC cell lines. TNBC cells were infected with *P. aeruginosa* at MOI 50, and viable intracellular bacteria were quantified by colony-forming unit (CFU) enumeration at serial time points following gentamicin treatment. CFU were measured at Days 0, 1, and 3 for all cell lines, with an additional Day 15 time point collected for MDA-MB-231 only. Values are expressed as log_10_(CFU+1) to enable visualization of time points with no detectable colonies. Data are presented as mean ± SD of two independent experiments. Clearance kinetics (Day 0–3) were analyzed by simple linear regression, with slopes compared between cell lines. Robust regression (Theil–Sen estimator) was also performed to confirm slope estimates.

**Figure 3 microorganisms-13-02317-f003:**
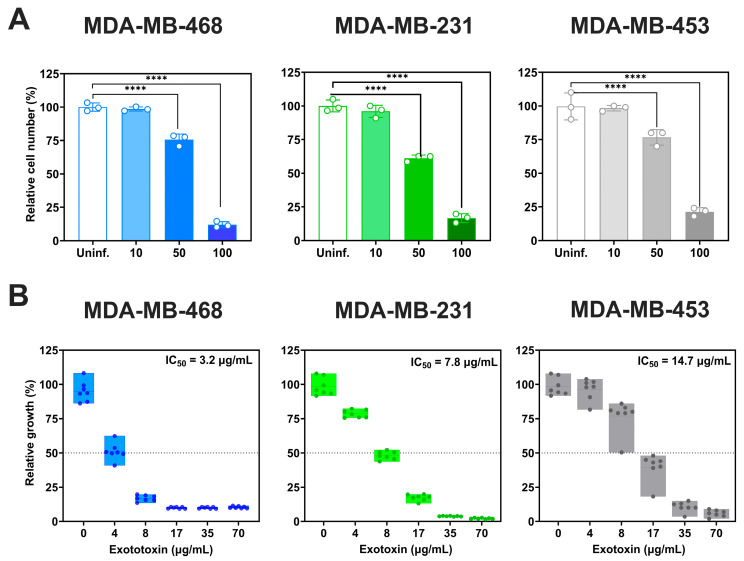
Cytotoxicity induced by *P. aeruginosa* infection and exotoxins in TNBC cell lines. (**A**) Acute cytotoxicity was assessed 2 h post-infection by determining the relative cell number, quantified as adherent biomass by crystal violet staining (OD_595_) and expressed relative to uninfected (Uninf.) controls. Data represent mean ± SD from at least three independent experiments. Statistical comparisons were performed by one-way ANOVA with Tukey’s multiple comparisons test against uninfected controls **** *p* < 0.0001). (**B**) Relative cell growth was determined by crystal violet staining 5 days post treatment with total *P. aeruginosa* exotoxins. Data were normalized to untreated controls (0 µg/mL exotoxin), and IC_50_ values were estimated by four-parameter logistic (4-PL) nonlinear regression. Data represent the mean ± SD from at least seven replicates.

**Figure 4 microorganisms-13-02317-f004:**
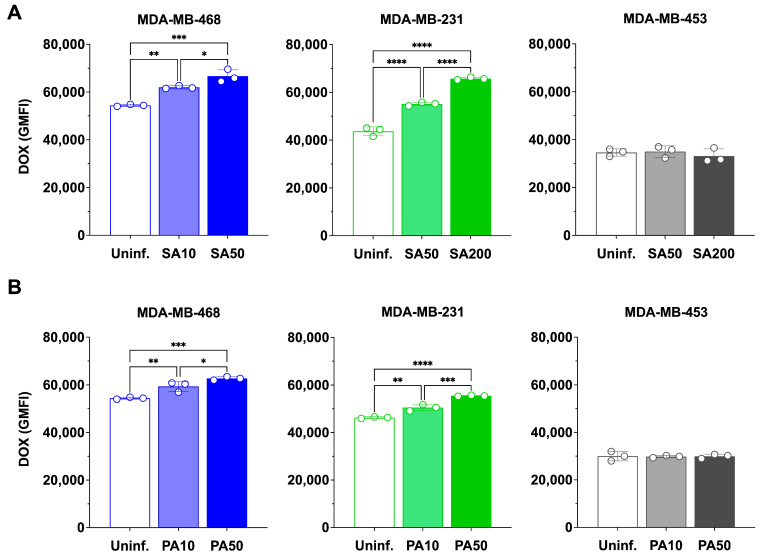
Effect of *S. aureus* and *P. aeruginosa* infection on doxorubicin accumulation in TNBC cell lines. (**A**) Cells were infected with *S. aureus* (SA) at MOIs of 10 and 50 (MDA-MB-468) or 50 and 200 (MDA-MB-231, MDA-MB-453). (**B**) Cells were infected with *P. aeruginosa* (PA) at MOIs of 10 and 50. After infection and removal of extracellular bacteria with gentamicin, cells were treated with 100 nM doxorubicin (DOX) for 24 h. Intracellular DOX accumulation was quantified in viable cells as geometric mean fluorescence intensity (GMFI) using spectral flow cytometry. Uninfected (Uninf.) DOX-treated cells served as controls. Data represent the mean ± SD from three independent experiments. Statistical significance was assessed by one-way ANOVA with Tukey’s multiple comparisons test (* *p* < 0.05; ** *p* < 0.01; *** *p* < 0.001; **** *p* < 0.0001).

**Figure 5 microorganisms-13-02317-f005:**
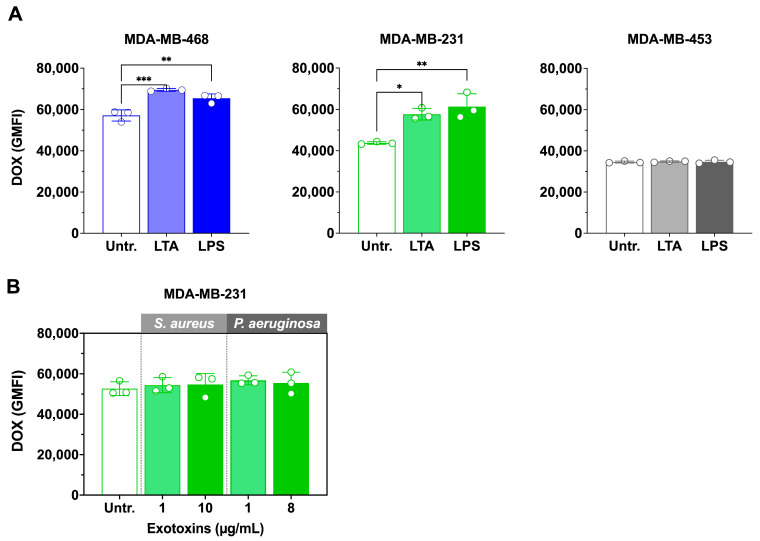
Effect of bacterial ligands and exotoxins on doxorubicin accumulation in TNBC cell lines. (**A**) MDA-MB-468, MDA-MB-231, and MDA-MB-453 cells were treated with 10 µg/mL *P. aeruginosa* lipopolysaccharide (LPS) or *S. aureus* lipoteichoic acid (LTA), together with 100 nM doxorubicin (DOX), for 24 h. (**B**) Total exotoxins *S. aureus* (SA Exo) and from *P. aeruginosa* (PA Exo) were prepared, and their IC_50_ values determined (10 µg/mL for SA Exo; 8 µg/mL for PA Exo). MDA-MB-231 cells were then treated with exotoxins at IC_50_ or at a lower concentration (1 µg/mL), together with 100 nM DOX, for 24 h. Intracellular DOX accumulation (GMFI) was quantified in viable cells by spectral flow cytometry. DOX-treated cells (no ligand or exotoxin) served as controls. Untreated (Untr.) Data are mean ± SD from three independent experiments. Statistical significance was assessed by one-way ANOVA with Tukey’s multiple comparisons test (* *p* < 0.05, ** *p* < 0.01; *** *p* < 0.001).

**Figure 6 microorganisms-13-02317-f006:**
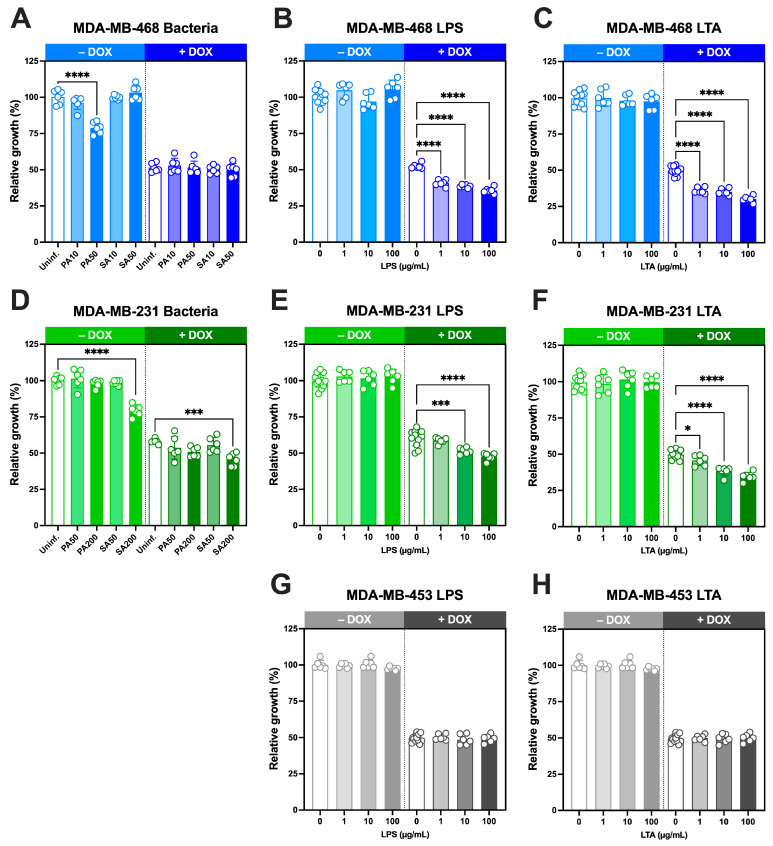
Effect of intracellular bacteria and bacterial ligands on TNBC cell growth in the presence or absence of doxorubicin. MDA-MB-468 (**A**–**C**, blue), MDA-MB-231 (**D**–**F**, green), and MDA-MB-453 (**G**,**H**, grey) cells were treated with (**A**,**D**) *P. aeruginosa* (PA) or *S. aureus* (SA) at the indicated multiplicities of infection (MOI), (**B**,**E**,**G**) *P. aeruginosa* lipopolysaccharide (LPS; 1–100 µg/mL), or (**C**,**F**,**H**) *S. aureus* lipoteichoic acid (LTA; 1–100 µg/mL) in the absence (– DOX) or presence (+ DOX) of doxorubicin (IC_50_ concentrations: 29.2 nM for MDA-MB-468, 37.7 nM for MDA-MB-231, 51 nM for MDA-MB-453). After 5 days, growth was quantified by crystal violet staining and expressed relative to uninfected (Uninf.) or untreated controls. Data are mean ± SD from 6–12 replicates. Statistical significance was determined using one-way ANOVA with Dunnett’s test, comparing treatments to untreated controls (– DOX) or DOX-alone controls (+ DOX). * *p* < 0.05, *** *p* < 0.001, **** *p* < 0.0001.

**Table 1 microorganisms-13-02317-t001:** Summary of *S. aureus* and *P. aeruginosa* in TNBC cell lines. Data for *S. aureus* are from [9]; *P. aeruginosa* data are from the present study. All measurements were taken at MOI 50, 24 h post-infection. eFluor 450 geometric mean fluorescence intensity (GMFI) reflects the relative intracellular bacterial load, and the percentage of eFluor 450-positive cells (% positive) indicates the proportion of infected cells. Data are presented as mean ± SD from 3 independent experiments.

	*S. aureus* % Positive	*P. aeruginosa* % Positive	*S. aureus* GMFI	*P. aeruginosa* GMFI
MDA-MB-468	97.6 ± 2.8%	12.5 ± 2.4%	49,5167 ± 35,000	4119 ± 56
MDA-MB-231	91.3 ± 3.8%	29.4 ± 3.6%	204,3410 ± 20,041	32,459 ± 2200
MDA-MB-453	13.4 ± 1.2%	0.3 ± 0.1%	9243 ± 230	203 ± 0

## Data Availability

The raw data supporting the conclusions of this article will be made available by the authors on request.

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
