# Peer review of "Bacterial Infections and Their Cell Wall Ligands Differentially Modulate Doxorubicin Sensitivity in Triple-Negative Breast Cancer Cells"

_microorganisms, 2025, doi:10.3390/microorganisms13102317_

Round 1

Reviewer 1 Report

Comments and Suggestions for Authors

This article is a well-designed study exploring Pseudomonas aeruginosa and Staphylococcus aureus and their cell wall components (LPS and LTA) for DOX accumulation and cytotoxicity on TNBC cells. Experiments are thoughtfully organized, data are well obtained, and findings strongly verify the authors’ principal result that bacterial infection and its corresponding ligands accelerate DOX accumulation and cytotoxicity, potentially by TLR2/4 signaling. This is a good study that reveals novel insights concerning host–microbe interactions at the tumor microenvironment with potential translational relevance.

However, there are a few issues that deserve additional explanation or elaboration, as follows, to fortify the mechanistic explanation as well as the translational vision of this paper.

  1. What cell did you use for figure 1D? Please indicate it.
  2. For figure 3A, I noticed that you obtained the viable cell number of MDA-MB-453 with infection of MOI 0, 10, 50, 100. Did you check the “internalization”and “persistence” with MOI 100 infection for these three cell lines?
  3. You tested the accumulation and the effect of dox on cell growth with both PA and SA, but did you confirm the “internalization”and “persistence” of SA in TNBC cells? Or please cite your previous published work.
  4. On line 371-372, “In MDA-MB-468 cells...whereas aeruginosa at MOI 50significantly reduced growth (Figure 6A).”, but the figure shows no significant changes.
  5. While your results strongly demonstrate that PA, SA, and their cell-wall components (LPS and LTA) enhanceDOX accumulation and cytotoxicity of TNBC cells, the extent of DOX accumulation depicted by Figures 4–5 appears relatively low. Does such a minimal increase of DOX accumulation yield the relatively low augmentation of cytotoxicity depicted by Figure 6? Furthermore, since LPS and LTA are potent stimulants of TLR2/TLR4 signaling pathways and are capable of inducing vigorous immune and inflammatory responses in vivo, do you anticipate (maybe in future work) that such effects—combining with increased DOX accumulation—would more strongly potentiate anti-tumor activity in animal models? In the meantime, since we already have known established toxicities of LPS and LTA, it would be better if you test safer TLR agonists or delivery systems could be considered as alternatives for reducing adverse effects while keeping anti-tumor activity.

Reviewer 2 Report

Comments and Suggestions for Authors

Having read the manuscript, my major concerns are as follows:

  1. Table 1. column 4 – S.aureus GMFI – please, correct the place for comma indicating thousand values for GMFI.
  2. Figure 1B and 1C – statistical analysis is incorrect. Please, perform a trend analysis for increasing MOI (5, 10, 50) for the respective cell lines independently. The Tukey’s post-hoc comparison of each value with every each value has no sense. Please, compare the MOI values for the same cell line, separately. Presentation of asterisks is not enough when presenting statistical significance. Please, add information about F-statistics, dfs and p values from one-way ANOVA test. Additionally, please add information about n – the analyzed items (samples). It was clearly stated that experiments were performed in triplicates, but no info about number of items/samples performed in one experiment. The data are presented as mean with SD values.
  3. Figure 2 – It is not clear why linear regression analysis was performed for these values. Since values were measured several times for the same cell lines, it would be better to analyze data with "one-way ANOVA with repeated measure on time". Since the experiments were performed in duplicates, please, add information about number of items/samples in one experiment. The data are presented as mean with SD values.
  4. Figures 4 and 5 – Please, add information about F-statistics, df and p values from one-way ANOVA test. Asterisks are not enough.

Round 2

Reviewer 2 Report

Comments and Suggestions for Authors

No further comments. All my previous comments have been explained.